# Pulmonary Embolism in Pregnancy: A Review for Clinical Practitioners

**DOI:** 10.3390/jcm13102863

**Published:** 2024-05-13

**Authors:** Agata Makowska, Thomas Treumann, Stefan Venturini, Michael Christ

**Affiliations:** 1Emergency Department, Cantonal Hospital Lucerne, 6000 Lucerne, Switzerland; stefan.venturini@luks.ch (S.V.); michael.christ@luks.ch (M.C.); 2Cardiology, Hospital Centre of Biel, 2501 Biel, Switzerland; 3Radiology, Cantonal Hospital Lucerne, 6000 Lucerne, Switzerland; thomas.treumann@luks.ch

**Keywords:** pulmonary embolism, pregnancy, multidisciplinary team, management, review

## Abstract

Diagnostic and therapeutic decision-making in pregnancy with suspected pulmonary embolism (PE) is challenging. European and other international professional societies have proposed various recommendations that are ambiguous, probably due to the unavailability of randomized controlled trials. In the following sections, we discuss the supporting diagnostic steps and treatments. We suggest a standardized diagnostic work-up in pregnant patients presenting with symptoms of PE to make evidence-based diagnostic and therapeutic decisions. We strongly recommend that clinical decisions on treatment in pregnant patients with intermediate- or high-risk pulmonary embolism should include a multidisciplinary team approach involving emergency physicians, pulmonologists, angiologist, cardiologists, thoracic and/or cardiovascular surgeons, radiologists, and obstetricians to choose a tailored management option including an interventional treatment. It is important to be aware of the differences among guidelines and to assess each case individually, considering the specific views of the different specialties. This review summarizes key concepts of the diagnostics and acute management of pregnant women with suspected PE that are supportive for the clinician on duty.

## 1. Introduction

Pulmonary embolism (PE), a major cause of maternal mortality in pregnant women, occurs in approximately 1 in 1000–3000 pregnancies [1]. Pregnancy is a hypercoagulable state caused by increased levels of fibrinogen, d-dimers, and coagulation factors (VII, VIII, and IX) with decreased protein S activity [2,3]. Not surprisingly, pregnancy-induced physiological adaptations are associated with significant changes in physiological measures, signs and symptoms, and laboratory test results that are comparable to those of pregnant patients with PE. Most clinical risk scores used to assess PE are not validated for pregnant women because of the limited number of relevant prospective studies. In recent years, European and other international societies have proposed various algorithms for the assessment of suspected PE in pregnancy [4,5,6,7,8,9,10,11]. The paucity of concise, evidence-based recommendations leads to different clinical practices [10]. In this review, we propose a practical summary of the important diagnostic steps and suggest a practical management concept of pregnant women with suspected PE at the time of presentation to the emergency department.

## 2. Pathophysiological Changes Contributing to PE during Pregnancy

PE is associated with complex changes during pregnancy. Detailed concepts are presented in Figure 1.

A major factor for increased risk of PE in pregnancy is related to Virchow’s triad: hypercoagulability, venous stasis and turbulence, and endothelial injury and dysfunction [12,13]. Moreover, pregnancy is a state of hypercoagulability due to changes in procoagulatory proteins: factors I, II, VII, VIII, IX and X increase during pregnancy. On the other hand, pregnancy increases resistance to antithrombotic factors such as protein C and protein S [14]. Thrombophilia can exacerbate these changes in clotting proteins, further increasing the patient’s risk of PE [15]. Moreover, excessive weight gain during pregnancy is associated with increased hypercoagulability and vascular changes that contribute to clot formation [16]. Inadequate nutrient intake, maternal age, and gestational diabetes mellitus (GDM) increase the risk of PE due to changes in insulin sensitivity and blood glucose levels, which may affect blood coagulation factors and increase the likelihood of thrombotembolic events [15,16].

Of note, placenta-related factors such as sFLT-1 (soluble FMS-like tyrosine kinase-1) have been suggested to increase the risk of PE. sFLT-1 plays a crucial role in thromboembolic diseases, particularly in the development of pre-eclampsia due to endothelial dysfunction [17]. Most thromboembolic events occur postpartum, because of vascular trauma during childbirth. Twins, triplets, or other multiple pregnancies are associated with a higher risk of PE due to increased blood volume and altered blood flow.

Risk factors modulated by the fetus may be caused by placental abruption and intrauterine growth restriction [16].

## 3. Clinical Presentation

The clinical manifestations of PE during pregnancy are variable and non-specific. The most common clinical manifestations of PE during pregnancy pertain to the respiratory system, but those respiratory signs and symptoms, such as dyspnea at rest or on exertion, pleuritic chest pain, cough, and tachypnea, are not specific for diagnosis.

Specific risk factors for pulmonary embolism during pregnancy include previous thromboembolism, age > 35 years, body mass index > 30 kg/m^2^, multiparity, comorbidities, pre-eclampsia, and immobility [18,19,20]. Of note, data supporting risk stratification based only on clinical risk factors are limited and uncertain [6,21,22,23]. The clinical manifestations of PE are comparable among pregnant women with PE, non-pregnant women with PE, and pregnant women without PE. To compare and better understanding we summarize the most of symptoms in mentioned groups in Table 1 [21,24,25,26,27].

There are some established scoring systems that predict PE. The well-established Wells and revised Geneva score can be used for calculating PE risk and guidance to perform a computed tomography pulmonary angiography (CTPA) in general population with suspicion of PE. One of Wells’s criteria reflects the clinical subjective opinion: “PE is the most probable diagnosis or equally likely”. For comparison, the PADUA score includes clinical features from medical history such as active cancer, previous VTE, reduced mobility, known thrombophilic conditions, recent trauma and/or surgery, elderly age, heart and/or respiratory failure, acute myocardial infarction and/or ischemic stroke, acute infection and/or rheumatologic disorder, obesity (BMI ≥ 30), and ongoing hormonal treatment. Each risk factor is assigned a certain number of points, and the total score determines the need for thromboprophylaxis in hospitalized, non-pregnant adult patients and is also an useful tool for stratifying patients before they undergo CTPA [28].

Tools to estimate the pretest probability for PE in the general population (Wells score, revised Geneva score, PADUA score, PERC rule and IMPROVE score) seem not to be valuable for use in pregnant women due to their non-optimal sensitivity, specificity, PPV, and NPV in pregnancy. For example, the Wells score has 40.7%, 81.5%, 44%, and 79.4%, respectively, in these categories, and the revised Geneva score has 62.9%, 59.2%, 35.4%, and 81.8%, respectively [5,7,10,29,30].

In 2021, the pregnancy-adapted Geneva (PAG) score was introduced to estimate the pretest probability of pregnant women with suspected PE. The PAG score appears promising and may facilitate medical decision-making at first contact [31]. The PAG score includes (1) age ≥ 40 years, (2) lower limb surgery or fracture in the last month, (3) previous deep vein thrombosis or PE, (4) unilateral lower limb pain, (5) hemoptysis, (6) tenderness of the lower limb and unilateral edema, and (7) heart rate > 110 bpm [31] (Table 2).

The PAG score ranges from 0 to 20 points. Patients were categorized as having low (0–1 points), medium (2–6 points), or high (≥7 points) clinical pretest probability [31], which corresponded to prevalence rates of 2.3%, 11.6%, and 61.5% of PE in pregnant patients [31]. This indicates that the PAG score may be valuable in predicting the prevalence of PE in group of pregnant women with suspected PE.

The PAG score shows a high discriminative power to identify patients at low, intermediate, or high pre-test probability (PTP) compared with the Geneva Score (AUC: 0.795 vs. AUC: 0.684 in ROC analysis, respectively). Nevertheless, prospective external validation of the PAG score is required [31]. Of interest, the YEARS algorithm has been prospectively examined, which was used in conjunction with d-dimer measurement and consisted of the following YEARS items: clinical signs of DVT, hemoptysis, or the most likely diagnosis of PE. This approach is discussed in the next section.

## 4. Laboratory Test Values

Physiological changes during pregnancy influence several laboratory markers [32]; d-dimer levels normally increase during pregnancy as gestation progresses [33,34,35]. In the first trimester, the d-dimer threshold is 50% higher, in the second trimester 100% higher, and in the third trimester, it is 125% higher than the normal threshold [30,36].

In this context, Kovac et al. suggested d-dimer thresholds for the first (286 ng/mL), second (457 ng/mL), and third trimesters of pregnancy (644 ng/mL) to exclude PE in this patient cohort [35]. In 2021, an Iranian research group suggested higher d-dimer cut-off values: a d-dimer cut-off value of 1447 g/L displayed a high specificity and sensitivity for PE diagnosis, irrespective of the gestational age [36]. In addition, the following reference ranges for the d-dimer level have been suggested: 169–1202, 393–3258, and 551–3333 µg/L for the first, second, and third trimesters of pregnancy, respectively [36]. All these data require prospective external validation. Of major interest, the following factors may influence d-dimer levels during pregnancy, regardless of the course of the pregnancy [37]:Type of pregnancy: in normal twin pregnancies, d-dimer levels are significantly higher during pregnancy than in normal singleton pregnancies [37].Gestational diabetes mellitus (GDM): the plasma d-dimer values of the GDM group are significantly higher in the third trimester than those of the group with normal singleton pregnancies [37].Arterial hypertension: hypertensive disorders in pregnancy constitute an independent risk factor for venous thromboembolism, which causes d-dimer level increments [38].Type of delivery: plasma d-dimer levels are significantly higher 24–48 h after delivery in women who underwent cesarean section than in women who gave birth vaginally [37].Breastfeeding: women who breastfeed have higher d-dimer levels [39].

An approach using a laboratory test (d-dimer) alone to exclude PE does not provide sufficient negative predictive value in pregnancy. Among other examples, the DiPEP study showed that d-dimer alone could not distinguish between pregnant and postpartum women who have PE and those who do not [22,23,40], particularly in women at high risk and/or in the third trimester of pregnancy (Table 3). Furthermore, it has been discussed not to overuse d-dimer measurements in pregnancy.

Finally, the combination of the clinical pretest probability (C-PTP) and laboratory tests increased the diagnostic value of risk prediction, as shown in the pregnancy-adapted YEARS approach (Figure 2).

The CT-PE pregnancy and ARTEMIS studies demonstrated the safety of excluding PE during pregnancy by a negative d-dimer test in cases of low or intermediate C-PTP [40,41,42,43] using the Geneva score and a pregnancy-adapted YEARS model. The YEARS algorithm integrates the following clinical findings: (1) signs of deep vein thrombosis, (2) hemoptysis, and (3) PE as the most likely diagnosis, which was evaluated based on the patient’s history and physical examination results. The thresholds for d-dimer levels were a cut-off of 500 µg/L used in patients with at least one item and a cut-off of 1000 µg/L in patients without any of those items [21,23,40,41,42]. The negative predictive value of the YEARS algorithm was highest during the first trimester of pregnancy and lowest during the third trimester. Despite the lower efficiency during this period of pregnancy, CTPA was avoided in 32% of cases [21,44]. In addition to these findings, the study by Kearon et al. found that in patients with a low C-PTP and a d-dimer level of <1000 ng/mL or a moderate C-PTP and a d-dimer level of <500 ng/mL, pulmonary embolism can be ruled out without further testing. Overall, these studies provide significant evidence in favor of the use of clinical probability-adjusted d-dimers in the management of PE [45].

## 5. Electrocardiogram

The electrocardiogram (ECG) has low sensitivity as a diagnostic test for PE, including pregnant patients [46]. Sometimes, changes may appear on an ECG that raise suspicion and support the diagnosis, such as sinus tachycardia, right bundle branch block, right axis deviation, or the classic S1Q3T3 pattern. However, even in the presence of a massive embolism, these changes may be absent. Moreover, physiological changes can occur during pregnancy that affect the ECG, such as left axial deviation and/or the presence of pronounced Q waves in ECG leads II, III, and aVF [46].

## 6. Imaging

Current discussions in clinical practice cover the question of which imaging modality should be used to exclude or diagnose PE in pregnancy. To facilitate clinical decisions, the challenges of different modalities will be discussed, such as specificity, sensitivity, negative predictive value, radiation dose, treatment, and diagnostic consequences. It is essential to mention that a d-dimer test, which is elevated in cases of DVT, may not always indicate the presence of PE. For stable patients, especially pregnant ones, avoiding a CT scan can prevent unnecessary exposure to radiation if it does not alter treatment plans.

### 6.1. Diagnostic Accuracy of Plain Chest Radiography

Chest X-ray is often recommended in the guidelines but appears not to be the imaging modality of choice for the diagnosis of PE in pregnant women. Chest radiographs often appear normal even in patients with PE [18]. The DiPEP study found that the presence of PE and non-PE-related abnormalities in chest X-rays increased the likelihood of a PE diagnosis [22,40]. In some guidelines, chest radiography is used to decide whether lung scintigraphy or CTPA should be used [7,10]. In addition, chest radiography may rule out non-PE-related diagnoses such as pneumothorax or pneumonia.

### 6.2. Diagnostic Accuracy of Echocardiography

The negative predictive value of bedside echocardiography is about 50%, indicating that echocardiography alone cannot be used to exclude the diagnosis of PE [6]. Notably, echocardiography is commonly available and can be used as a “rule-in” test in the emergency department at the patient’s bedside, supporting the decision for immediate treatment in high-risk PE [47,48,49]. For example, the McConnell sign (akinesia of the midportion of the right ventricular wall with preserved right ventricular function of the apex) and reduced longitudinal systolic function (TAPSE) show 100% specificity [50]. Right ventricular overload signs (D-shaping, dilatation of the right ventricle) have a sensitivity of approximately 80% for the diagnosis of PE [50], supporting the usefulness of echocardiography, especially in hemodynamically unstable patients with suspected PE [6]. The use of speckle-tracking echo (STE)—especially RV strain—appears promising to assess the severity of RV dysfunction also in pregnant women with PE. While there is extensive research on its benefits in non-pregnant individuals, its application among pregnant women, especially those with suspicion of PE, is still an area of ongoing study [51].

### 6.3. Diagnostic Value of Bilateral Venous Compression Ultrasound of the Lower Extremities

Compression ultrasonography (CUS) is useful in pregnant women with lower limb pain or swelling; therefore, it is incorporated into the pregnancy-adapted algorithms of the YEARS algorithm (ARTEMIS study) [21]. The use of the pregnancy-adapted YEARS algorithm excludes PE in 80% of pregnant women without the need for radiation exposure [19,41]. In the CT-PE-Pregnancy study, CUS was performed on 75% of the overall study population: among patients without leg symptoms, 2% were diagnosed with proximal deep vein thrombosis (DVT), and 9% of patients with leg symptoms had more positive results [41].

However, isolated deep pelvic vein thrombosis is a common cause of false-negative CUS results [52]. This limitation was already considered in the second consensus document on the diagnosis and treatment of acute deep vein thrombosis in pregnant women [53]. Thus, CUS should also include the visualization of the iliac veins or indirect signs of pelvic thrombosis, such as the monophasic flow of the common femoral vein [54,55]. The effectiveness of CUS without considering the clinical context (a sign of thrombosis) is limited [54,56]. It should be interpreted by including C-PTP and/or calf swelling with a diameter ≥ 3 cm greater than that of the asymptomatic calf [53,57]. The more sensitive method is magnetic resonance venography (sensitivity 92%, specificity 99%) and should be performed as the next diagnostic step if an isolated deep vein thrombosis in the pelvis is suspected but cannot be safely ruled out using ultrasonography due to the low sensitivity of 50% (the specificity was 99%) [58]. Some case reports have described the usefulness of magnetic resonance direct imaging (MRDTI) in the diagnosis of isolated venous thrombosis [59].

### 6.4. Diagnostic Value of Lung Scintigraphy

Several international guidelines and prospective studies [4,60] recommend the following diagnostic steps based on the GRADE (grades of recommendation, assessment, development) system: chest X-ray (CXR) as the first radiation-associated procedure, perfusion scintigraphy of the lung as the preferred test for normal CX, and CTPA over digital subtraction angiography (DSA) for non-diagnostic ventilation-perfusion (V/Q) results [61]. Perfusion scintigraphy of the lung displays a diagnostic accuracy comparable to CTPA, with different rates of non-diagnostic results: 5.9–14% vs. 4–12% respectively [62,63]. This difference is probably because there is no accepted definition of “non-diagnostic results” in the Cochrane database. The median negative predictive value of lung scintigraphy was 100% [62]. To summarize, lung perfusion scintigraphy is suitable for ruling out PE during pregnancy if the chest X-ray (CXR) is normal [4,61].

### 6.5. Diagnostic Value of Computed Tomographic Pulmonary Angiography

In the Cochrane database, the median negative predictive value for CTPA was 100%, and the median sensitivity was 83% [62]. The negative predictive value in pregnant women is high (100%), probably because of the low prevalence of PE and the low clinical probability of predisposed pregnant patients [64,65]. Additional reviews from 2018 and 2019 confirmed that CTPA is suitable for ruling out PE during pregnancy and that the diagnosis is based on a validated YEARS algorithm [62,63]. However, it is necessary to be aware of some methodological limitations, such as poor pulmonary arterial opacification and artifacts due to respiratory motion, which are more common in pregnant women than in non-pregnant ones [66]. Approximately 4–36% of CTPAs in pregnant women are non-diagnostic [18,62,63,66,67,68]. Thus, modifications of the imaging protocol are required to improve image quality. Some modifications, such as high concentration, high volume, and high rate of contrast injection followed by saline flush or shallow inspiration breathing, appear to be useful [69]; however, they are standard in the meantime with the use of modern CT scanners [70]. Close collaboration between the physician ordering the examination and the radiologist is of major importance.

### 6.6. Diagnostic Value of Magnetic Resonance Imaging

Recently, new techniques have been developed to improve spatial resolution, reduce the acquisition time, and reduce motion artifacts in magnetic resonance imaging for the diagnosis of pulmonary embolism. Herèdia et al. investigated balanced steady-state free-precession imaging. The protocol visualizes central, lobar, and segmental arteries with sufficient image quality in pregnant women [71]. Perfusion magnetic resonance imaging is the best stand-alone technique for PE diagnosis in pregnant women (sensitivity and specificity of 100% and 91%, respectively) [72]. According to the American College of Obstetricians and Gynecologists (ACOG), the use of gadolinium contrast agents in MRI should be restricted during pregnancy. Gadolinium may only be used as a contrast agent in a pregnant woman if it significantly improves diagnostic performance and is expected to improve fetal or maternal outcomes. Breastfeeding should not be interrupted after gadolinium administration [73].

Studies suggest that non-contrast MRI (with free-breathing arterial spin labeling) is a viable alternative for the diagnosis of pulmonary embolism [74]. A recent study showed that non-contrast magnetic resonance angiography (MRA) has high sensitivity and specificity in the diagnosis of pulmonary embolism, particularly in the proximal pulmonary arteries [75]. Another study discussed the sensitivity and specificity of non-contrast MRI and contrast-enhanced MRA. Non-contrast MRI had 89% sensitivity and 98% specificity, whereas contrast-enhanced MRA had 81% sensitivity and 100% specificity [76]. Mudge et al. reported the feasibility of diagnosing PE using non-contrast MRI. Although their MRI protocol was not optimized for the detection of pulmonary embolism, it was still 69% sensitive per vessel and 82% sensitive per patient [76]. Nevertheless, MRI is not currently the gold standard, and further research and evaluation are needed to validate its clinical utility. MRI techniques cannot be used in routine clinical practice for the diagnostic work-up of suspected pulmonary embolism in pregnant and non-pregnant women because of a lack of outcome studies demonstrating necessary safety and feasibility.

### 6.7. Radiation- and Contrast-Enhanced Computed Tomographic Pulmonary Angiography versus Pulmonary Scintigraphy

The fetal radiation dose is low in both CTPA and lung scintigraphy, with a mean dose of 0.01–0.66 mGy for CTPA and 0.1–0.8 mGy for V/Q-SPECT protocols [77]. Of interest, the radiation dose is higher in the third trimester than in the first trimester [78]. The radiation doses for 256-slice CTPA are listed in the online Appendix A [61]. Moreover, CTPA may cause a 0.2–2.2% increased the relative lifetime risk of breast or lung cancer in young mothers [79]. The radiosensitivity of the breast in pregnant women is higher than that in other parts of the body and in non-pregnant women because of several factors, including increased cell division, hormonal changes, and increased blood flow [80].

Strategies to minimize the absorbed radiation dose include the use of bismuth shields, breathing strategies, and automatic exposure controls [79]. An ongoing prospective multicenter study (OPTICA study) will evaluate the efficacy and safety of a low-dose CTPA protocol [81]. Of note, many studies have shown that V/Q scanning produces lower effective doses to the breast and fetus than CTPA [82,83,84]. In CTPA and perfusion scintigraphy, the average doses were estimated to be effective doses of 21 mGy and 1.04 mGy, doses absorbed by the maternal breast of 10–70 and 0.22–0.28 mGy (per breast), and doses absorbed by the uterus and fetus of 0.46 mGy and 0.25 mGy, respectively [84]. Despite these limitations, the risks associated with radiation exposure for both CTPA and V/Q scanning appear to be lower than the risk of a missed PE diagnosis [18], and CT angiography is the preferred method of diagnosis, as suggested by several guidelines [61]. In addition, several techniques exist to reduce the radiation dose (for example, photon-counting computed tomography) by approximately 48% and still have high sensitivity, such as a reduced scan length defined by the upper part of the aortic arch and the upper part of the lower hemidiaphragm [85,86]. It is important to realize that scintigraphy is only available to a limited extent in daily clinical practice. CTPA is readily available in most hospitals. This advantage usually influences decisions in daily clinical practice and outweighs the consideration of radiation exposure in urgent cases. The iodinated contrast agents used during CTPA were classified as pregnancy category B safe drugs by the Food and Drug Administration (FDA) [87].

In summary, the use of contrast media and radiation for diagnostic imaging in pregnant women with suspected pulmonary embolism, is considered safe when used prudently and when the benefits of accurate diagnosis and management outweigh the minimal risks. According to the American College of Obstetricians and Gynecologists (ACOG), the radiation exposure from CT scans is much lower than the dose associated with fetal harm, and the use of iodinated contrast media has not been shown to harm the fetus [88].

### 6.8. Diagnostic Value of Lung Sonography

Lung ultrasonography (LUS), a point-of-care diagnostic tool, is particularly beneficial in the diagnosis of critically ill patients, for whom transport to a radiology department may entail additional risks. It facilitates the detection of parenchymal changes in the lungs that indicate pulmonary embolism. In addition, an integrated approach with triple point-of-care ultrasonography, which simultaneously examines the lungs, heart, and leg veins, may improve the diagnostic accuracy of PE. This method could be particularly beneficial for pregnant women because it reduces radiation exposure and bed rest [86]. Unfortunately, the use of lung sonography to detect PE in pregnant women is currently not established. Studies performed using the BLUE protocol have shown 81% sensitivity and 99% specificity for the diagnosis of pulmonary embolism in patients with acute dyspnea in the presence of an “A-profile” and a deep vein thrombosis (DVT) [89].

The distribution of lesions detected by LUS is shown in Figure 3 [90]. It is evident that several pulmonary emboli involve the peripheral lung. Furthermore, it is evident that central PE with hemodynamic relevance cannot be detected by LUS with sufficient sensitivity due to air-filled lungs around the central structures, causing total reflection of ultrasound waves. An autopsy study of lethal PE reported multiple pulmonary infarcts (peripheral lesions) of different ages in up to 15% of affected patients [91]. Hence, central PE with hemodynamic relevance could be better diagnosed using echocardiography.

The typical characteristics of suspicious (peripheral) lesions are [92]:Hypoechoic, pleural-based parenchymal lesion, usually wedge-shaped (>85%), sometimes round or polygonal, may be present.A central hyperechoic lesion may be present in 20% of cases, indicating an air-filled bronchiole.Lesions may be associated with pleural effusion.Color Doppler cannot detect pulmonary arterial flow in pulmonary infarction. This is referred to as “consolidation with low perfusion”.A congested thromboembolic vessel may be visible, which is referred to as a “vascular sign”.The posterior lower parts of the lung are affected in most patients (>70%). Although the explanation for this is not clear, it could be due to the anatomical structure of the pulmonary tree. In addition, the posterior lower parts of the lung are one of the easiest to access by transthoracic ultrasonography.The right lung is more frequently affected than the left one (66.7%).

In a prospective multicenter study based on triangular, hypoechoic, and pleural parenchymal lesions, LUS showed a 74% sensitivity and 95% specificity for the diagnosis of peripheral PE (positive predictive value: 95%, negative predictive value: 75% and accuracy: 84%) [92].

LUS is a non-invasive, widely available, cost-effective method that can be performed rapidly. A negative LUS examination cannot rule out PE, but a positive LUS finding with moderate/high suspicion of PE can prove to be a valuable tool in the diagnosis of PE [90].

In international consensus, LUS is recognized as an alternative diagnostic tool for the diagnosis of PE when CT is contraindicated, not available, or declined by the patient (level of evidence: A) [93]. Multi-organ ultrasonography (heart, lung, and leg veins) increases the accuracy of clinical pre-test probability estimation in patients with suspected PE, and with 90% sensitivity and 86.2% specificity. Thus, use of those multiple ultrasound techniques may reduce the indication of multidetector computed tomography pulmonary angiography (MDCTPA) with associated ionizing radiation exposure [94]. To date, three societies (ATS-STR, ESC, RCOG) recommend deep vein ultrasonography before further imaging if lower limb symptoms are present, while two others recommend ultrasonography regardless of the presence of clinical manifestations of deep vein thrombosis (GTH, SOGC) [10].

In summary, the diagnostic role of LUS is currently not well defined and not recommended to exclude PE. Nevertheless, LUS may contribute specific information in specific groups of patients.

## 7. Therapy

The acute treatment of PE depends on the risk group of the affected patient. In stable patients (very low/low/and low–intermediate risk groups), systemic anticoagulation is the first line of treatment. In patients with hemodynamic instability, invasive support with thrombolytic therapy or interventional approaches such as catheter-directed therapy or surgical thrombectomy should be considered. We describe the different treatment options for the affected pregnant patient.

### 7.1. Anticoagulation

Low molecular weight heparin (LMWH) and unfractionated heparin (UFH), which are used as first-line anticoagulation therapies, are safe during pregnancy and breastfeeding because they do not cross the placenta [2,77]. UFH can be administered both intravenously and subcutaneously. PTT in response to UFH is shortened in pregnant women [95]. However, improved monitoring can be achieved by analyzing anti-factor Xa levels during UFH use (target values: 0.5–0.80 IU/mL). Current guidelines do not recommend routine anti-factor Xa level monitoring in pregnant women with normal renal function during LMWH administration [77]. Anti-factor Xa levels should be monitored in patients with mechanical heart valves, impaired renal function, extreme body weight (<50 kg and >90 kg), or recurrent VTE despite anticoagulation [11]. If dose adjustment is required, the peak serum anti-factor Xa level should be measured four hours after injection. No data are available on dose adjustment during pregnancy [96]. The use of new oral anticoagulants is contraindicated during pregnancy because they can cross the placenta and affect the blood coagulation system of the fetus and also exert teratogenic effects [97]. LMWH should be discontinued 12–24 h before lumbar instrumentation (such as epidural placement) or cesarean delivery and at least six hours before vaginal delivery. The restart of LMWH should be discussed in a multidisciplinary team. Usually, anticoagulation is started within six hours after vaginal delivery or within twelve hours after a cesarean section [11,77].

### 7.2. Thrombolytic Treatment

There is a paucity of evidence for the safety and efficacy of thrombolytic therapy in pregnant women. Systemic thrombolysis is only recommended in unstable high-risk patients [6] and is associated with favorable maternal outcomes in patients with massive PE; the maternal survival rate was approximately 92% [98]. Complications such as cardiac arrest (17.6%) and severe maternal hemorrhage (28.4%) occur during systemic thrombolysis. The fetal mortality rate is up to 20% [98]. Alteplase is usually administered at a dose of 100 mg over two hours, so the likelihood of transplacental transition is low [99]. Intravenous UFH administration should be initiated immediately after thrombolysis. At our institution, pregnant women receiving thrombolytic therapy are closely monitored by our obstetric team, and all treatments are discussed during interdisciplinary rounds.

### 7.3. Catheter-Directed Therapy (CDT)

CDT is intended for high-risk patients in whom thrombolysis or appropriately dosed anticoagulation has either failed or is contraindicated [100]. The rate of major hemorrhage (including intracranial hemorrhage) is low (18%) [101]. There are already more than ten devices used to treat unstable non-pregnant patients, which are based on aspiration, mechanical fragmentation, and/or thrombolytic infusion [100]. To date, CDT has rarely been used in affected pregnant patients with PE. The maternal survival rate was 100% in reported cases of mechanical percutaneous thrombectomy based on thrombus aspiration, thrombus fragmentation, and rheolytic thrombectomy [102,103]. There is a paucity of randomized trials comparing these treatments with systemic thrombolysis, even in the non-pregnant population. Therefore, the decision to perform percutaneous thrombolysis and/or thrombectomy should be individually discussed, including factors such as a high risk of bleeding, expected radiation exposure, and the availability of the procedure. We recommend that CDT procedures should be performed only in specialized centers [86,99].

### 7.4. Surgical Thrombectomy

Surgical thrombectomy is more frequently considered for the treatment of PE during pregnancy, especially in cases where anticoagulation is ineffective, or the patient is hemodynamically unstable [104]. Surgical thrombectomy with cardiopulmonary bypass is performed without cardioplegia by removing proximal pulmonary clots through incisions in the two main pulmonary arteries. Hemodynamic improvement and/or the return of spontaneous circulation was documented in 93.8% of patients after surgical thrombectomy and maternal survival occurred in 84–86.1% of patients [98].

### 7.5. Extracorporeal Membrane Oxygenation

Venoarterial extracorporeal membrane oxygenation (VA-ECMO) is considered a lifesaver for patients with high-risk PE [99,105]. Venous and arterial cannulas are placed in the inferior vena cava and common femoral artery, respectively. VA-ECMO is usually combined with fibrinolysis or embolectomy to improve outcomes of affected patients. The 30-day mortality rate for patients with high-risk PE treated with and without ECMO was 62% and 43%, respectively [106]. The most appropriate indication for VA-ECMO is refractory cardiac arrest due to PE. Only a few case reports have been published in which these invasive techniques have been used as rescue strategies [99,105,106,107]. There are studies that recommend ECMO with thrombolysis or surgical thrombectomy. However, it is unclear which trial is superior to another [108] in the general population, even in pregnant women (there are no known studies in this group).

## 8. Follow-Up

Follow-up care after an acute PE during pregnancy involves several important steps [109]:Identify optimal anticoagulation strategies after delivery.Age-appropriate screening for cancer.Regular exercise and sports activities.Ensure adherence to anticoagulants and avoid relevant drug interactions.Exclude chronic thromboembolic pulmonary hypertension in all patients with persistent clinical manifestations of dyspnea or right heart failure.Assess the overall bleeding risk.

Treatment of pregnancy-associated PE should be continued for at least three months, including six weeks postpartum [96,97]. The discussion about anticoagulant therapy duration in pregnant women is controversial due to the paucity of available data. Hormonal contraceptives (such as birth control pills) can be continued during anticoagulant treatment to prevent pregnancy and mitigate the risk of abnormal uterine bleeding. Current evidence suggests that there is no increased risk of recurrent PE in women receiving combined hormonal or progestin-only contraceptives during anticoagulation [109].

## 9. Conclusions

Diagnostic and therapeutic decision-making in pregnant patients with suspected PE is challenging. Therefore, we suggest that the treatment of affected patients should involve a multidisciplinary team approach with emergency physicians, pulmonologists, angiologist, cardiologists, thoracic surgeons, radiologists, and/or obstetricians as team members to choose optimal diagnostic and therapeutic approaches. The pulmonary embolism response team (PERT) has been suggested to improve the outcomes of care for high-risk PE patients [110]. Unfortunately, evidence from prospective cohort or interventional studies is rare, and recommendations for the diagnosis and management of PE in this situation are either ambiguous or contradictory in currently available guidelines. 

Questions for future research

Diagnostic role of lung ultrasonography for the safe rule-out of PE in pregnant patients.Decision support for the initial diagnosis of PE in the emergency department (Figure 4).Combinations of laboratory tests (including cardiac biomarkers) to increase the sensitivity of the diagnostic algorithm for PE in pregnancy.Summarized take home messages are presented in online Appendix A.

## Figures and Tables

**Figure 1 jcm-13-02863-f001:**
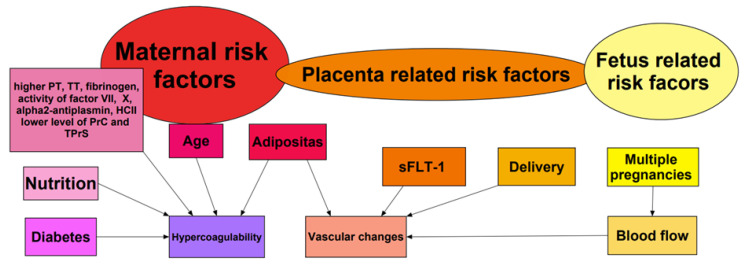
Pathophysiological changes and factors contributing to PE during pregnancy. PT—prothrombin time; TT—thrombin time; HCII—activity of heparin cofactor II; PrC—protein C; TPrS—total protein S; sFLT-1—soluble FMS-like tyrosine kinase-1.

**Figure 2 jcm-13-02863-f002:**
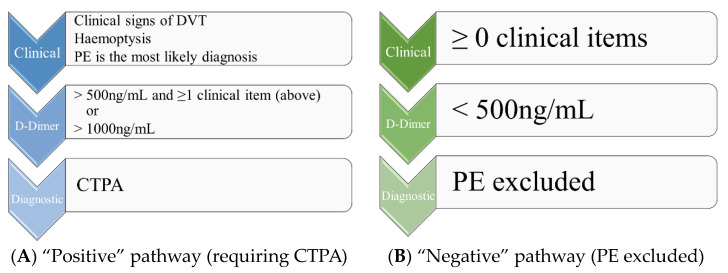
(**A**,**B**) Simplified pregnancy-adapted YEARS algorithm for the management of suspected acute pulmonary embolism in pregnant patients (modified from van der Pol et al. [21]). DVT, deep vein thrombosis; PE, pulmonary embolism; CTPA, computed tomography–pulmonary angiography.

**Figure 3 jcm-13-02863-f003:**
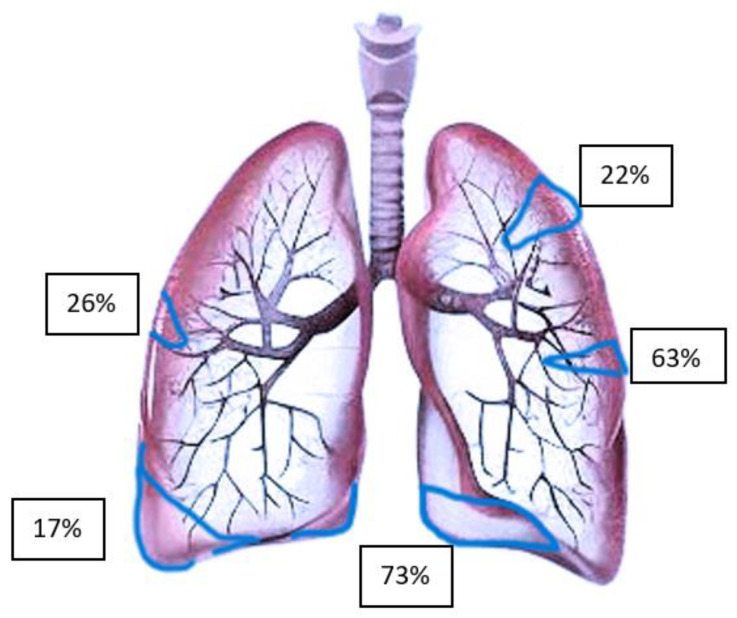
Distribution of pulmonary embolism lesions detected by transthoracic ultrasonography (Adapted from Comert et al. [90]).

**Figure 4 jcm-13-02863-f004:**
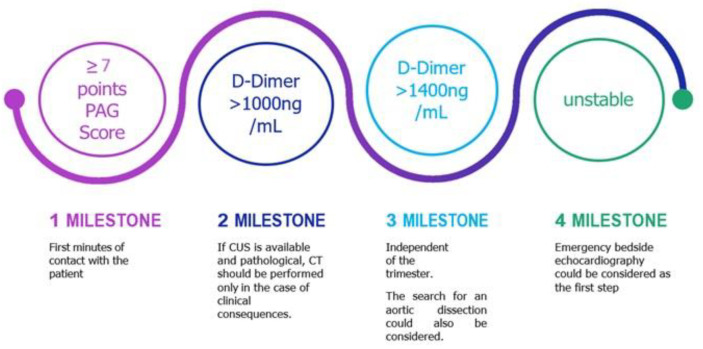
Proposed fast track CTPA evaluations for emergency physicians.

**Table 1 jcm-13-02863-t001:** Comparison of clinical manifestations of PE in different patient populations.

Symptoms	Pregnant Women with PE [%]	Pregnant Women without PE [%]	General Population with PE [%]
Dyspnea	62	60–75	66–97
Chest pain	46 (pleuritic)	93	28–45
	19 (nonpleuritic)		
Hemoptysis	8	No data	6–16
Signs or symptoms of lower limb DVT	7	1	38–55
Syncope	No data	No data	7–38

PE—pulmonary embolism; DVT—deep vein thrombosis.

**Table 2 jcm-13-02863-t002:** The Pregnancy-Adapted Geneva score (modified from Robert-Ebadi et al. [31]).

Pregnancy-Adapted Geneva Score		
**ITEM**		**POINTS**
Age ≥ 40 years		+1
Surgery (under GA) or lower limb fracture in the past month		+2
Previous DVT or PE		+3
Unilateral lower limb pain		+3
Hemoptysis		+2
Lower limb tenderness and unilateral edema		+4
Heart rate > 110 bpm		+5
Maximum point number		20
**Points**	**Category**	**PE Prevalence**
0–1	Low	1.0–4.9%
2–6	Intermediate	6.9–18.9%
≥7	High	35.5–82.2%

GA, general anesthesia; DVT, deep vein thrombosis; PE, pulmonary embolism.

**Table 3 jcm-13-02863-t003:** Comparison of the DiPEP, ARTEMIS, and CT-PE-Pregnancy studies.

Study	Study Design	Used C-PTP	Recommendations and Key Findings
DiPEP	- prospective and retrospective, descriptive - women during and after pregnancy	- the PERC rule, Well’s PE criteria, and the simplified revised Geneva score	- clinical decision rules and blood tests alone should not be used to determine suspected PE in pregnancy or postpartum. - d-dimer and other biomarkers were not reliable in ruling out PE during pregnancy without a clinical context [23]
ARTEMIS	- international, multicenter - prospective management study - pregnant women	- 3 YEARS criteria: 1. clinical signs of deep vein thrombosis 2. hemoptysis 3. pulmonary embolism is the most likely diagnosis	- the YEARS algorithm is safe and effective, and it results in less radiation exposure compared with conventional diagnostic methods - d-dimer and other biomarkers were not reliable in ruling out VTE during pregnancy without a clinical context
CT-PE- Pregnancy	- prospective study - pregnant women or postpartum women	- the revised Geneva score [41]	- the YEARS algorithm safely excludes PE in pregnant women and reduces the need for CTPA [19]

DiPEP: an observational study of the diagnostic accuracy of clinical assessment, d-dimer, and chest radiography for suspected pulmonary embolism in pregnancy and postpartum; C-PTP: clinical pretest probability; PERC rule: pulmonary embolism rule-out criteria.

## Data Availability

Not applicable.

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
