# Peer review of "Pulmonary Embolism in Pregnancy: A Review for Clinical Practitioners"

_jcm, 2024, doi:10.3390/jcm13102863_

Round 1

Reviewer 1 Report

Comments and Suggestions for Authors

Authors successfully summarized knowledge on diagnostic and treatment of pulmonary embolism in pregnant women.

There is a great issue with diagnostic and treatment of pulmonary embolism in pregnant women.

Authors did a narrative review based on the PRISMA-Equity 2020 expansion of the published literature on pregnant women with suspected acute PE. They searched the PubMed, EMBASE, Google Scholar, Web of Science, and Cochrane Library databases for relevant full publications addressing the diagnosis and treatment.

The title is adequate and will attract the attention of the readers.

Abstract is comprehensive and adequate. Introduction is written very well and we have all the relevant information regarding the review conducted.

In the Materials and Methods part authors gave a good description of the used methodology.

Clinical presentation of PE in pregnancy could be very various and pre-test probability test have positive and negative issues. All of this is very well presented in the parts numbered as 3,4 and 5.

Imaging diagnostic is explained in details with the take a home messages. All literature used is contemporary. Lung ultrasonography is also mentioned and it will be more used in clinical practice in the future as well as a MRI without contrast. Authors explained that we need further validation of this methods for diagnosis.

Regarding the therapy of PE in pregnancy all available treatments are mentioned, even an ECMO in hemodynamically compromised patients.

Authors even mentioned follow up of those patients.

Conclusion part is fine. We need more precise diagnostic algorithms (with avoidance of radiation exposition as much as possible) and measurement of D dimer is very uncertain in pregnancy or puerperium so it should be cautiously be interpreted.

Author Response

Dear Reviewer,
Thank you for taking the time to review our manuscript and for providing valuable feedback.

Reviewer 2 Report

Comments and Suggestions for Authors

In this "narrative" review, Dr. Makowska and colleagues discussed pulmonary embolism in pregnancy, focusing on the discrepancies between guidelines and the recommendations to tackle the problem. Overall, although this manuscript holds a potential, I do have several concerns to address:

1) In line 13, the authors declared that this is a narrative review. However, they wrote methods section and seemed to performed a systematic review by partially adhering to PRISMA guideline and searching through databases. Which one is correct? A narrative review is not done systematically. Please revise the manuscript according to the appropriate design. 

2) Although this was aimed for clinicians, the readers would benefit from pathophysiological explanation of PE in pregnancy. Hypercoagulable state is not the only cause of PE in pregnancy. Some cases of thromboembolic events are related to the release of sFLT1 in placenta (PMID: 38395115). Please discuss the pathophysiological aspects in a separate section. 

3) Also, please add an illustrative figure capturing the complex pathophysiology of PE in pregnancy

4) "Clinical manifestations of deep vein thrombosis are calf or thigh pain with or without swelling, erythema, edema, and tenderness" why did the authors include DVT as well? I thought this manuscript discusses PE only? Please clarify

5) In the clinical presentation section, the authors must explain the differences of symptoms/signs of PE in pregnancy vs. non-pregnant women or perhaps general population. 

6) Please clarify whether PAG score predicts PE prevalence or PE risk?

7) Are there any comparison between PAG score and other established scoring systems such as PADUA or Wells score? Please add into the discussion.

8) Based on the text, I concluded that D-dimer is useless to identify PE in pregnancy, is this correct? If so, the authors need to be bold and suggest not to check it. If there is a room for D-dimer in pregnancy, the authors need to discuss it. Please comment on this.

9) Line 157 indicates that the readers should use YEARS instead of PERC. So why bother mentioning PERC in the manuscript? Please clarify.

10) Also, based on that statement, D-dimer is still useful despite all the potential bias related to normal pregnancy. How to mitigate the false positive results?

11) Please be consistent. If the authors want to show the take home messages for each section, they need to show them for all sections. Otherwise, they can move the recommendations at the end of the manuscript and make one big table from them. 

12) Is there any benefits of speckle-tracking echo? it has been discussed at least in non-pregnant women

13) Please also address the risk of contrast media and radiation caused by CT and chest x-ray. Is the benefit outweigh the risk?

14) I don't see the benefit of LUS in PE except if the lesions are in the peripheral border of the lung, which is uncommon for PE. Can the authors comment on this? How often are PE found adjacent to the lung border?

15) "Follow-up" should be in a separate section rather than inside ECMO section. Please correct the error. 

16) I am not sure if I understand Figure 3. What does it aim at? Please clarify and revise. First, it needs caption. Second, what does it mean by 1 milestone, 2 milestone etc? Then, this recommendation is different than the ones above (line 157). Please be consistent. 

Comments on the Quality of English Language

Some (minor) typographical errors are present.

Round 2

Reviewer 2 Report

Comments and Suggestions for Authors

Thanks for the response, I do have some additional comments:

1) please remove the material section, it is not necessary for a narrative review

2) "Due to the same 75 pathomechanism, These condition are summarized as “venous thromboembolism (VTE)". We do understand this, but the title is PE, so the discussion should be limited to PE. Otherwise, the title, abstract and content of this manuscript should be changed into VTE instead of PE.

Comments on the Quality of English Language

No comment

Author Response

Please see the attachment,
